# Plasminogen Activator Inhibitor-1 (*PAI-1*) Gene Polymorphisms Associated with Cardiovascular Risk Factors Involved in Cerebral Venous Sinus Thrombosis

**DOI:** 10.3390/metabo11050266

**Published:** 2021-04-23

**Authors:** Anca Elena Gogu, Andrei Gheorghe Motoc, Alina Zorina Stroe, Any Docu Axelerad, Daniel Docu Axelerad, Ligia Petrica, Dragos Catalin Jianu

**Affiliations:** 1Department of Neurology, Victor Babes University of Medicine and Pharmacy, 300041 Timisoara, Romania; agogu@yahoo.com (A.E.G.); dcjianu@yahoo.com (D.C.J.); 2Department of Anatomy, Victor Babes University of Medicine and Pharmacy, 300041 Timisoara, Romania; amotoc@umft.ro; 3Department of Neurology, General Medicine Faculty, Ovidius University, 900470 Constanta, Romania; docuaxi@yahoo.com; 4Department of Kinetotherapy, Brainaxy Clinic, 900628 Constanta, Romania; docuaxy@yahoo.com; 5Department of Internal Medicine II, Division of Nephrology, Victor Babes University of Medicine and Pharmacy, 300041 Timisoara, Romania; petrica.ligia@umft.ro; 6Centre for Molecular Research in Nephrology and Vascular Disease, Faculty of Medicine, Victor Babeș University of Medicine and Pharmacy, 300041 Timișoara, Romania; 7Centre for Cognitive Research in Neuropsychiatric Pathology (Neuropsy-Cog), Faculty of Medicine, Victor Babeș University of Medicine and Pharmacy, 300041 Timișoara, Romania

**Keywords:** cerebral venous sinus thrombosis, *PAI-1* gene polymorphisms, homocysteine

## Abstract

Cerebral venous sinus thrombosis (CVST), accounting for less than 1% of stroke cases, is characterized by various causes, heterogeneous clinical presentation and different outcome. The plasminogen activator inhibitor-1 (*PAI-1*) gene polymorphisms has been found to be associated with CVST. The aim of this retrospective study was to determine the potential association of *PAI-1 675 4G/5G* polymorphisms and homocysteine levels with cardiovascular risk factors in a group of young patients with CVST. Eighty patients with CVST and an equal number of age and sex matched controls were enrolled. The protocol included demographic and clinical baseline characteristics, neuroimagistic aspects, genetic testing (*PAI-1 675 4G/5G* polymorphisms), biochemical evaluation (homocysteine—tHcy, the lipid profile, blood glucose, glycohemoglobin—HbA1c, high-sensitive C-reactive protein—hsCRP) data, therapy and prognosis. The *PAI-1 675 4G/5G* gene polymorphisms were significantly correlated with increased homocysteine level (tHcy) (*p* < 0.05), higher total cholesterol (TC) (*p* < 0.05), low- density lipoprotein cholesterol (LDLc) (*p* = 0.05) and high- sensitive C- reactive protein (hsCRP) (*p* < 0.05) in patients with CVST when compared with controls. From the PAI-1 gene polymorphisms, the PAI-1 675 4G/5G genotype presented statistically significant values regarding the comparisons of the blood lipids values between the CVST group and control group. The homocysteine (tHcy) was increased in both groups, patients versus controls, in cases with the homozygous variant *4G/4G* but the level was much higher in the group with CVST (50.56 µmol/L vs. 20.22 µmol/L; *p* = 0.03). The most common clinical presentation was headache (91.25%), followed by seizures (43.75%) and focal motor deficits (37.5%). The superior sagittal sinus (SSS) was the most commonly involved dural sinus (56.25%), followed by the lateral sinus (LS) (28.75%). Intima—media thickness (IMT) values were higher in the patients’ group with CVST (0.95 mm vs. 0.88 mm; *p* < 0.05). The fatal outcome occurred 2.5% of the time. *PAI-1 675 4G/5G* gene polymorphisms and higher homocysteine concentrations were found to be significantly associated with CVST in young patients.

## 1. Introduction

Cerebral venous sinus thrombosis (CVST) accounting for less 1% of stroke cases is characterized by various causes, heterogeneous clinical presentation and different outcome [1]. Among multifactorial causes of CVST, which can be of infective (otitis media, mastoiditis, sinusitis, infections of the midface, meningitis, systemic infections with Staphylococcus aureus, fungus, Herpes zoster virus, Human immunodeficiency virus, etc.) or non- infective nature (head trauma, brain tumors, neurosurgical procedures, pregnancy and puerperium, consuming of oral contraceptive drugs and asparaginase therapy, inflammatory multisystem diseases, etc.), prothrombotic state (thrombophilias) stands out, it’s diagnosis is based on highly sophisticated tests [2,3]. In the International Study on Cerebral Vein and Dural Sinus Thrombosis (ISCVT) inherited and acquired thrombophilia includes protein C or S deficiency, antithrombin deficiency, factor V Leiden mutation, prothrombin *G20210A* mutation, *MTHFR* gene mutation, hyperhomocysteinemia [4,5]. Genetic polymorphisms of plasminogen activator inhibitor *PAI-1 675 4G/5G* was lately correlated with CVST. Plasminogen activator inhibitor type 1 is an endogenous glycoprotein from the group of serine proteases, which is produced in endothelial cells, liver, etc. and plays an important role in the regulation of fibrinolysis [6]. *PAI-1*synthesis is regulated by the gene on chromosome 7 in whose promoter region at position—675 bp mutation is described which results in heterozygous variant of *4G/5G* but *4G* only transcriptionally active allelic variants responsible for increases risk of thrombosis [6]. In the most cases the contribution of *PAI-14G/5G* polymorphism of the synergetic conditioned by the presence of the conventional risk factors for the occurrence of arterial and venous thrombosis, such as hypertension, diabetic disease, and chronic inflammatory diseases which can have a significant impact on the level of *PAI-1*in plasma [7,8]. An association has also been observed between the *4G* allele and cholesterol (TC) and low-density lipoprotein (LDL) plasma levels in patients with coronary artery disease [9]. In the present study we explored the association of *PAI-1 675 4G/5G* polymorphisms and homocysteine levels with cardiovascular risk factors in a group of young patients with CVST.

## 2. Results

### 2.1. Patient Population

#### 2.1.1. Demographic Characteristics

Eighty patients with CVST and an equal number of age and sex matched controls were enrolled in the study. In the CVST group there were 54 females and 36 males (55% vs. 45%). The mean age of the patients was 33.77 years (standard deviation Std = 6.60; range = 18–45 years). In the controls group we had the same number of females and males, but the mean age was 33.12 years (standard deviation Std = 6.12; range = 18–45 years). The demographics, clinical, neuroradiological and biochemical characteristics of the patients and controls from the study are summarized in the Table 1.

#### 2.1.2. Clinical Features

The clinical spectrum of CVST varies according to the disease etiology, the venous channel involved, age and the time interval between onset of disease and clinical presentation. Onset of symptoms was acute in 22 patients (27.5%), subacute in 50 patients (62.5%) and chronic in 10 patients (10%). The most common presentation was gradual headache (73 cases; 91.25%), focal or generalized seizures (35 cases; 43.75%), predominantly among patients with focal deficits or focal edema, ischemic and hemorrhagic changes on MRI of the brain. Focal neurological deficits such as hemiparesis, hemisensory disturbances, hemianopsia, aphasia, cranial nerve palsies occur in 30 cases (37.5%). Coma at onset was a predictor for death (2 cases: 2.5%).

In the CVST group we found 21 cases (26.25%) with hypertension and 13 cases (16.25%) in the control group (*p* < 0.005). The mean value of the systolic blood pressure was 128.56 mmHg (Std, 23.15) in the CVST group and 123 mmHg (Std, 19.03) in the control group; there are no significant difference between two groups (*p* = 0.075). The mean value of the diastolic blood pressure was 81.18 mmHg (Std, 14.78) versus control group, 75.37 mmHg (Std, 13.58) (*p* = 0.132).

#### 2.1.3. Neuroimaging

A native computed tomography (CT) scan was made at every patient on admission in hospital. The delta sign and the corn sign, classic signs of intracranial venous sinus thrombosis, appears on contrast enhanced CT at 22 patients (27.5%). Some of them had hemorrhagic infarcts (30 cases: 37.5%). In the first 48 h after admission, we performed MRI with diffusion-weighted imaging in combination with MR venography at all patients. An abnormal hyperintense signal in a thrombosed sinus on T1-weighted and T2-weighted MRI corresponding with absence of flow on MR venography is diagnostic [1]. The most common sites of thrombosis were the superior sagittal sinus (45 cases; 56.25%), the lateral sinus (23 cases: 28.75%) and cavernous sinus (12 cases; 15%). Cerebral edema was recognized at 52 patients (65%) and subarachnoid hemorrhage at 8 patients (10%).

We describe next the most special cases, having a various clinical symptomatology, correlated with neuroradiological findings.

The first case was a female, 28 years old with an etiology of thrombophlebitis em-phasized by presence of *PAI-1* gene polymorphism-4G/4G genotype; she was hospi-talized with severe psychomotor agitation, generalized seizures, slight hemiparesis. During the evolution, after a period of six months, as a consequence of the cerebral post thrombotic syndrome and edematous encephalopathy, the same patient presented residual symptomatology: generalized seizures with a frequency of 3–4 episodes per month under anticonvulsant treatment and psychical disturbances (irritability, insomnia). The imag-istic features can be seen in Figure 1 and Figure 2.

The second case was a female, 24 years old with an etiology of thrombophlebitis accentuated by the presence of *PAI-1* gene polymorphism-homozygous phenotype 4G/4G and oral contraceptives use, hospitalized with meningeal symptoms, left hemiparesis, generalized seizures and psychomotor agitation. During the evolution, after six months, the patient did not show any residual symptomatology related to thrombophlebitis. The image features can be seen in Figure 3 and Figure 4.

Extracranial Doppler ultrasound was performed in all patients and in the control group for detected carotid atheromathosis with intima- media thickness (IMT) measurements. We want to make the evaluation of cardiovascular risk factor; mean IMT = 0.93 mm (Std, 0.17) in the CVST group and mean IMT = 0.91 mm (Std, 0.16) in the control group with *p*-value < 0.05.

#### 2.1.4. Biochemical and Genetic Findings

When neuroimaging studies reveal CVST, the next step is to determine the cause and investigation should be individualized. Because it is common for patients to have more than one risk factor for CVST we detailed biochemical and genetic testing associated with cardiovascular risk factors.

*PAI-1 675 4G/5G* gene and *MTHFR* gene seems to be involved in lipid metabolism [10]; we considered it important to detect the total cholesterol (TC) in both groups: mean value = 204.8 mg/dL (Std = 36.6) in the CVST group versus mean value = 198.26 mg/dL (Std = 34.11) in the control group; *p*-value < 0.05. Low-density lipoprotein cholesterol (LDLc) was 117.11 mg/dL (Std = 25.89) in the CVST group and 108.11 mg/dL (Std = 26.21) in the control group; *p*-value = 0.05. High-density lipoprotein cholesterol (HDLc) was found mean value = 56.65 mg/dL (Std = 15.19) versus mean value = 54.6 mg/dL (Std = 12.14) in the control group; *p*-value = 0.242. Triglycerides (TGL) was mean value = 144.15 (Std = 70.41) in the CVST group versus mean value = 131.33 mg/dL (Std = 67.73) in the control group; *p*-value < 0.05.

Blood glucose and glycohemoglobin (HbA1c) were interpreted according with American Diabetes Association (ADA). We found at blood glucose mean value = 102.07 mg/dL (Std = 34.93) in the CVST group and mean value = 93.91 mg/dL (Std = 24.56) in the control group; *p*-value = 0.084. Glycohemoglobin (HbA1c) was mean value = 5.56% (Std = 1.35) in CVST group and mean value = 5.24% (Std = 0.95) in the control group; *p*-value = 0.703.

Inflammatory markers are beneficial in the early stages of cerebral venous sinus thrombosis [11]. High-sensitive C-reactive protein (hsCRP) was determined in the CVST group with mean value = 7.67 mg/L (Std = 2.94) and in the control group was mean value = 4.39 mg/L (Std = 1.7); *p*-value, 0.05.

Some studies have suggested, in addition to the traditional risk factors, the existence of metabolic, haemostatic and genetic risk markers are important for coronary artery disease and, also, for CVST [12,13]. Relations have been found between increased baseline concentration of homocysteine and *PAI-1 675 4G/5G* gene polymorphisms in coronary and cerebral thrombotic events [14,15]. Homocysteine (tHcy) concentrations were significantly higher in patients than in controls. The mean value was 31.39 µmol/L (Std = 31.25) in the CVST group and mean value = 12.3 µmol/L (Std = 7.41) in the control group; *p*-value < 0.05.

Genotyping for *PAI-1 675 4G/5G* gene polymorphism showed a prevalence rate of 32 (40%) and 22 (27.5%) for *4G/4G* genotype in patients and control population, which is significantly different (*p*-value < 0.05). We observed a higher frequency of *4G/4G* genotype and a lower frequency of the *5G/5G* genotype in young patients with CVST compared to healthy controls. We also looked for the association of *4G/5G* polymorphism with homocysteine (tHcy) levels in the CVST group versus the controls group. These results show that *4G/4G* genotype has a significant association with high levels of homocysteine in both groups: 50.56 µmol/L (Std = 37.98) in CVST group and 20.22 µmol/L (Std = 7.41) in the control group; *p*-value = 0.032. In the CVST group, homocysteine has the highest level at *4G/4G* genotype, but it also has high levels at *5G/5G* (Table 2).

In our comparison of the CVST group and control group taking into consideration the *PAI-1* gene polymorphism, regarding the levels of the TC, LDLc, HDLc, TGL, hsCRP levels and regarding the IMT we have found statistical significance regarding the following:in the *5G/5G* genotype comparison was a statistically significant difference between the TGL levels (*p* < 0.001) between the CVST group and control group. The differences between the TC levels were not statistically significant (195.5 mg/dL vs. 194.2 mg/dL; *p* = 0.712) between the CVST group and control group. The differences between the LDLc levels were not statistically significant (116 mg/dL vs. 110.4 mg/dL; *p* = 0.124) between the CVST group and control group. The differences between the hsCRP levels were not statistically significant (6.61 mg/L vs. 4.34 mg/L; *p* = 0.398) between the CVST group and control group. The IMT values were not statistically significant between the CVST group and control group (0.85 mm vs. 1 mm; *p* = 0.966).in the *4G/4G* genotype comparison was not found a statistically significant difference between the TC, LDLc, TGL, hsCRP levels and IMT between the CVST group and control group. The differences between the TC levels were not statistically significant (199.03 mg/dL vs. 203.68 mg/dL; *p* = 0.201) between the CVST group and control group. The differences between the LDLc levels were not statistically significant (116.03 mg/dL vs. 114.23 mg/dL; *p* = 0.571) between the CVST group and control group. The differences between the TGL levels were not statistically significant (149.78 mg/dL vs. 153.32 mg/dL; *p* = 0.264) between the CVST group and control group. The differences between the hsCRP levels were not statistically significant (7.42 mg/L vs. 4.84 mg/L; *p* = 0.571) between the CVST group and control group. The IMT values were not statistically significant between the CVST group and control group (0.98 mm vs. 0.91 mm; *p* = 0.984).in the *4G/5G* genotype comparison was a statistically significant difference between the TC, LDLc and TGL levels (*p* < 0.001) between the CVST group and control group. The differences between the hsCRP levels were not statistically significant (8.02 mg/L vs. 4.17 mg/L; *p* = 0.264) between the CVST group and control group. The IMT values were not statistically significant between the CVST group and control group (0.9 mm vs. 0.86 mm; *p* = 0.991).

## 3. Discussion

Demographic, clinical and neuroradiologic presentation of CVST in our study was similar with the conclusions provided by other studies on the same matter, with increased frequency of causes which included thrombophilia. *PAI-1*gene polymorphisms have a high prevalence in CVST.

In our study, we found a young female predominance (55% versus 45%) and the mean age of the patients was 33.77 years. The symptomatology was varied according with etiology of the disease, location of the sinus thrombosis and comorbidities. The most common presentation was headache, followed by focal or generalized seizures and focal neurological deficit. In the CVST group, we found 21 cases with hypertension versus 13 cases in the control group, with significant differences (*p*-value < 0.05).

The most common sites of CVST in our study were the superior sagittal sinus, the lateral sinus and cavernous sinus. On MRI, cerebral edema was recognized at 52 patients and subarachnoid hemorrhage at 8 patients. In all patients and in the control group we had performed extracranial Doppler ultrasound for evaluation the carotid atheromatosis with intima-media thickness (IMT) measurements as cardiovascular risk factor. There were significant differences between the CVST group and the control group, mean rank of IMT = 0.95 mm (Std, 0.17) in the CVST and mean rank of IMT = 0.88mm (Std, 0.16) in the control group.

Our study aimed to find the associations of *PAI-1 675 4G/5G* gene polymorphisms, increased homocysteine levels, lipoprotein plasma concentration (total cholesterol, low-density lipoprotein cholesterol, triglycerides) and high-sensitive C-reactive protein (hsCRP). Data from the literature have documented that *PAI-1*, homocysteine and lipoprotein act as acute phase reactants in CVST [16,17,18,19,20,21,22]. In addition, these markers may be new potential pharmacological targets [22,23,24].

*PAI-1 675 4G/5G* seems to be involved in the lipid metabolism. In the present study, we found a significantly increased values for TC (*p*-value < 0.05), LDLc (*p*-value = 0.05), TGL (*p*-value < 0.05) in the group of patients with CVST. There was no association with blood glucose, HbA1c and patients with CVST and *PAI-1*gene mutation. In the *PAI-1 675 5G/5G* comparison was a statistically significant difference in the TGL levels (*p* < 0.001) between CVST and control groups. In the *PAI-1 675 4G/4G* comparison was not found a statistically significant difference in the TC, TGL, LDLc, HDLc and hsCRP levels between CVST and control groups. In the *PAI-1 675 4G/5G* comparison was a statistically significant difference in the TC, LDLc and TGL levels (*p* < 0.001) between CVST and control groups.

Interestingly, in the *PAI-1 675 4G/5G* comparison there was no statistically significant difference between the CVST group and the control group regarding the levels of tHcy, but statistical significances regarding the blood lipids levels were found. Furthermore, in the case of the *PAI-1 675 5G/5G* group, differences between the CVST and the control group regarding the blood lipids were not encountered, but we did find a statistically significant difference regarding levels of tHcy; one explanation for this encounter would be offered by the fact that CVST *PAI-1 675 5G/5G* patients numbered only six. Regarding the IMT findings in the *PAI-1* gene polymorphism in the CVST group and controls group were not found any statistically significant differences.

Inflammatory markers as high-sensitive C-reactive protein (hsCRP) was determined in the CVST group with mean value = 7.67 mg/L and in the control group was mean value = 4.39 mg/L (*p*-value < 0.05).

We and other studies have previously shown that homocysteine (tHcy) is an independent risk factor for CVST with etiology like inherited thrombophilia. Chen et al., managed to find the association between neuroinflammation due to overrated microglia activation, augmented by elevated plasma level of tHcy in a rat model with ischemic stroke [25,26,27]. Homocysteine concentrations were significantly higher in patients than in controls. We found the mean value = 31.39 µmol/L in the CVST group and mean value = 12.3 µmol/L in the control group; *p*-value < 0.05. 

The *PAI-1*gene plays a predominant role in the regulation of fibrinolysis and an overexpressive *PAI-1*may promote the occurrence of thrombotic events [28]. In this study we looked for *4G/5G* polymorphisms and homocysteine levels as a risk factor for CVST. Genotyping for *PAI-1 675*
*4G/5G* gene polymorphisms showed a prevalence rate of *4G/4G* genotype in patients and control population with significantly differences (*p*-value < 0.05). Also, we observed a higher frequency of *4G/4G* genotype (40%) and a lower frequency of *5G/5G* genotype (7.5%) in the CVST group compared with the control group (27.5% for *4G/4G* and 25% for the *5G/5G* genotype). 

Our present study has shown a significant association of the *4G* allele with CVST, and we also looked for the association of *4G/5G* polymorphism with homocysteine (tHcy) levels involved in the etiology of CVST. The results show that individuals carrying homozygous *4G* alleles have the highest mean of tHcy levels (50.56 µmol/L) in patients with CVST and 20.22 µmol/L in controls (*p*-value = 0.032). The individuals who were carrying homozygous 5G alleles from the CVST group have also the mean of tHcy levels higher than controls (21.5 µmol/L versus 11.1 µmol/L; *p*-value = 0.001).

In the studies of Varga et al., and Moll et al., was described that the people with a *MTHFR* genetic variant have a deficit in processing folate [28]. This defective gene will generate increased levels of homocysteine in the persons who inherit *MTHFR* variants from both parents [28,29]. Also, in the meta-analysis of Kelly et al., was concluded that mild-to moderate increased levels of Hcy are associated with ischemic stroke and also, the *MTHFR* genotype can possess an influence in the susceptibility to ischemic stroke [30]. *PAI-1* and homocysteine are involved in neurological brain pathology and Marcurcci et al. [13] wrote about the occurrence of major adverse cardiac events after successful coronary stenting as being related to *PAI-1* and homocysteine.

The limitation of our study was the retrospective nature and the small sample size with the recruitment of all patients from a single department. The similar designs must be carried out in large multicentric population. In addition, another limitation of the study was that we cannot be evaluated the plasma *PAI-1* levels, we made only genotyping the *4G/5G* polymorphism. Some studies have suggested that *PAI-14G/5G* polymorphism did not significantly affect *PAI-1*concentration measured at the time of the acute event; in this situation the influence of the inflammatory state on *PAI-1* concentration is scarcely genetically modulated [31].

## 4. Materials and Methods

### 4.1. Patient Population

This is a retrospective study including 80 young adults (18–45 years), 54 females and 36 males which were recruited from the Neurology Department of Timisoara County Emergency Clinical Hospital. Patients (*n* = 80) presenting with cerebral venous and sinus thrombosis were clinically and radiologically (computed tomography, magnetic resonance imaging—MRI, magnetic resonance venography—MRV) diagnosed were referred for the study. In the study were included patients diagnosed with comorbidities such as hypertension, type II diabetes mellitus, carotid atheromatosis and dyslipidemia. The exclusion criteria were: brain tumors, head trauma, oncologic pathologies, psychiatric diseases, hemorrhagic stroke, severe hematological disorders and treatment with oral anticoagulation. The control group comprised 80 healthy participants, aged and sex- matched from the same demographic area. The control population had to fulfill a questionnaire to ascertain that they were stroke-free [32]. The study was approved by Ethics Committee for Clinical Studies of the Timisoara County Emergency Clinical Hospital (registration number: 228/24.02.2021) and was conducted in accordance with the Declaration of Helsinki. The study participants both patients and controls gave a written informed consent for enrollment.

### 4.2. Clinical, Biochemical and Genetic Evaluation

The evaluation of general and neurological clinical condition was made at all patients on admission. The patients were considered to have hypertension diagnosed according to the European Guidelines on Cardiovascular Disease Prevention in Clinical Practice and were taking antihypertensive drugs [33]. Dyslipidemia was defined according to the report of the National Cholesterol Education Program and diabetes mellitus (T2DM) was defined in agreement with American Diabetes Association [34,35]. The blood samples were collected for each patient on admission and the results were provided by Medical Laboratory of Timisoara County Emergencies Clinical Hospital. Total cholesterol (TC), low-density lipoprotein cholesterol (LDLc), high- density lipoprotein cholesterol (HDLc), triglycerides (TGL), blood glucose, glycohemoglobin (HbA1c) were determined at all patients and the control group. The value of glycohemoglobin (HbA1c) higher than 6.5% was interpreted as T2DM. High-sensitive C-reactive protein (hsCRP) had a reference interval < 5 mg/L. Homocysteine concentration was measured by an immunoassay method (fluorescence polarization immunoassay, IMX system; Abbot) in plasma samples obtained after centrifuging blood collected into containing EDTA. Higher levels of homocysteine are split into three categories: moderate 15–30 µmol/L; intermediate 30–100 µmol/L; severe- greater than 100 µmol/L [36,37]. Genetic testing was performed at Bioclinica Laboratories (Timisoara, Romania). For detection of *PAI-1 675 4G/5G* polymorphism, genomic DNA was extracted from peripheral blood leucocytes using QIAamp DNA mini blood kits (QIAGEN, Hilden, Germany). The QIAamp DNA Blood Mini Kit facilitates the segregation of DNA from blood with fast spin-column or vacuum processes. The QIAamp DNA Blood Mini Kit operates sample sizes of up to 200 µL, with a preparation time of 20–40 min. The usual yield from 200 µL healthy whole blood is 4–12 µg, with an elution volume of 50–200 µL. The QIAamp DNA Blood Mini Kit process is automatable. PCR amplification was made using LightCycler 480 Instrument II Platform (Roche Diagnostics GmbH, Mannheim, Germany).

### 4.3. Neuroimaging

Computed tomography (CT) was performed for every patient on admission to the hospital. CT is useful in showing hemorrhagic infarction, subarachnoid hemorrhage or intraparenchymal hemorrhage resulting from CVST [1,38]. The delta sign and the cord sign are classic signs of CVST, appearing on contrast-enhanced CT. The magnetic resonance imaging (MRI) and magnetic resonance venography (MRV) was made for every patient in the first 48 h of admission. An abnormal hyperintense signal in a thrombosed sinus on T1-weighted and T2-weighted MRI corresponding with absence of flow on venography is diagnostic [1,39]. MR-angiography (three-dimensional time-of-flight 3D-TOF sequences) and MR-venography (2D-TOF) were obtained during the same imaging session and were interpreted by radiologists at the Timisoara County Emergencies Clinical Hospital [36].

Extracranial Doppler Ultrasound of carotid arteries was performed in all patients and in controls included in the study. In order to detect carotid atheromatosis, we measured intima-media thickness (IMT), value of <0.9 mm considered as normal.

### 4.4. Statistical Analyses

Statistical calculations were performed using Microsoft Excel (Microsoft Corporation, Albuquerque, NM, USA, version 2016). The Chi-square test of independence (also known as the Pearson Chi-square test, or simply the Chi-square) was used for testing hypotheses with nominal variables. Chi-square test compares the size any discrepancies between the expected results and the actual results, given the size of the sample and the number of variables in the relationship.

The data used in the described article are hypothesis driven research.

Statistical significance is covered by different observation regarding the presented study. *p*-value < 0.05 was used for significant differences. The statistical software generated the *p*-value and allowed the appreciation of the range of values in different contexts.

The Fisher-exact *p*-value corresponds to the proportion of values of the test statistic in which the differences are less than 5 for groups compared. Null hypothesis in this context was computed by considering all possible values that could give the tables observed.

Wilcoxon–Mann–Whitney two-sample rank-sum test was used to assess whether the distributions of observations obtained between two separate groups on a dependent variable are systematically different from one another.

## 5. Conclusions

*PAI-1 675 4G/5G* gene polymorphism seems to be involved in CVST etiology. The *PAI-1*gene mutation, especially the prevalence of *4G/4G* genotype was correlated with higher total cholesterol (TC), low-density lipoprotein cholesterol (LDLc), triglycerides (TGL), homocysteine level (tHcy) and high-sensitive C-reactive protein (hsCRP) in our study. Determining the thrombophilic risk profile, in particular *PAI-1 675 4G/5G* gene polymorphism and hyperhomocysteinemia of patients with CVST at the time of the acute event may allow for the selection of patients who will need follow up after hospital discharge.

## Figures and Tables

**Figure 1 metabolites-11-00266-f001:**
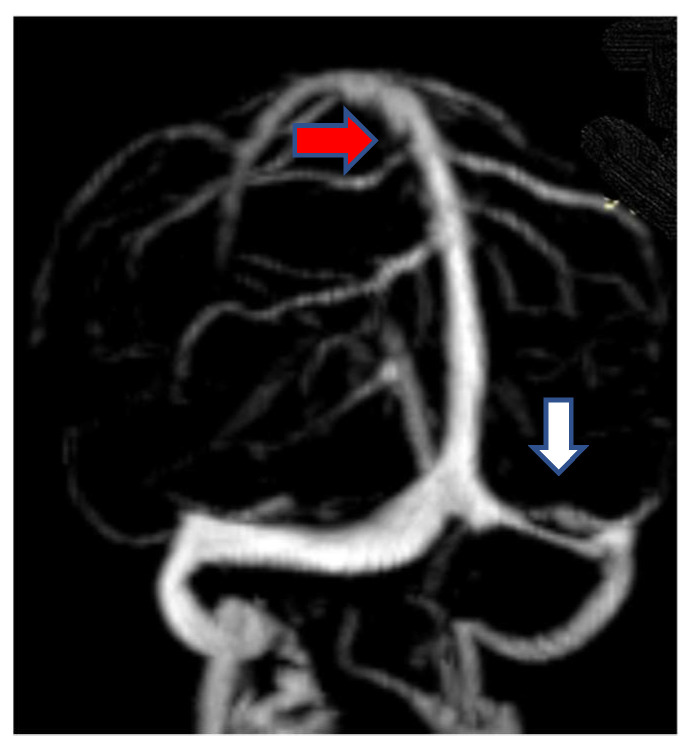
MR-venography Maximum Intensity Projection (MIP) range reveals thrombosis of the superior sagittal sinus (red arrow) and the left lateral sinus (white arrow) at onset.

**Figure 2 metabolites-11-00266-f002:**
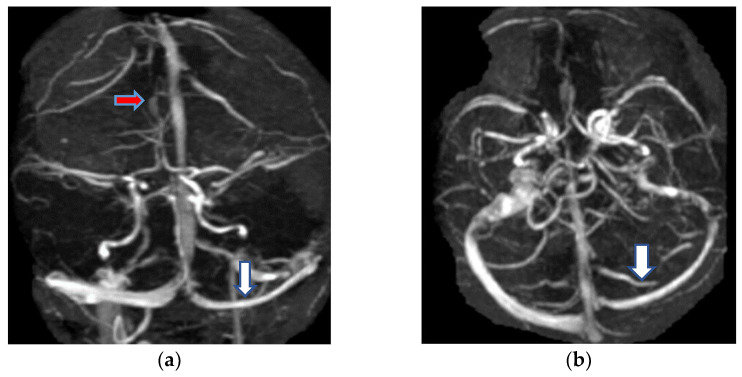
MR-venography 2D-TOF (2D-time of flight) in the coronal (**a**) and axial (**b**) plane reveals the permeabilization of the superior sagittal sinus (red arrow) and the left lateral sinus (white arrow) after six months of the onset.

**Figure 3 metabolites-11-00266-f003:**
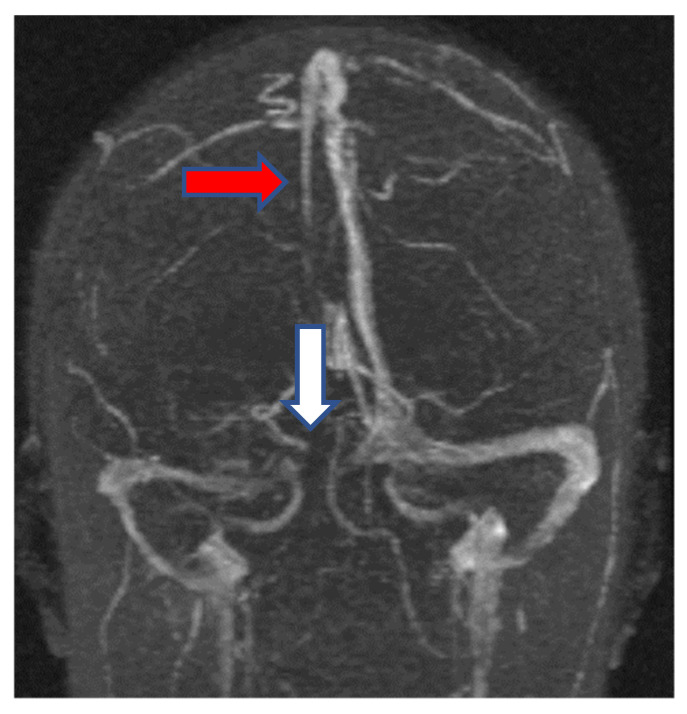
MR-venography 2D-TOF (2D-time of flight) in the coronal plane noting the absence of the flow in the superior sagittal sinus (red arrow) and the right lateral sinus (white arrow) with thrombosis of both sinuses at the onset.

**Figure 4 metabolites-11-00266-f004:**
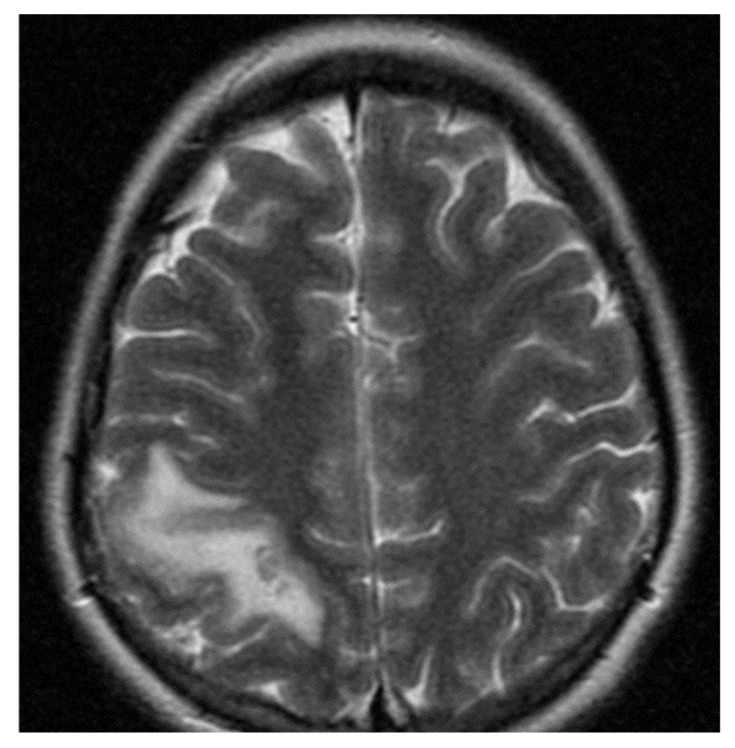
Magnetic resonance (MR) T2-weighted imaging in the axial plane shows cerebral infarction in the right parietal cortex crossing anatomic arterial distributions with hemorrhagic transformation.

**Table 1 metabolites-11-00266-t001:** Demographic data, clinical baseline, neuroradiological findings and biochemical characteristics of the study participants using the Mann- Whitney U Test, Chi Square Test and Fisher Test.

No.	Variable	CVST Group (*n* = 80)		Controls (*n* = 80)		*p*-Value
		Mean ± Std Deviation (Median)	Mean Rank	Mean ± Std Deviation (Median)	Mean Rank	
1	Age, years	33.77 ± 6.60	34.82	33.12 ± 6.12	32.04	0.374 *
2	Male *n* (%)	36 (45%)	37.31	36 (45%)	34.67	0.360 *
3	Female *n* (%)	54 (55%)	55.96	54 (55%)	52.01	0.263 *
4	HTN ^1^ *n* (%)	21 (26.25%)	19.61	13 (16.25%)	14.37	*p* < 0.05
5	SBP ^2^, mmHg	128.56 ± 23.15	131.79	123 ± 19.03	119.75	0.075
6	DBP ^3^, mmHg	81.18 ± 14.78	82.57	75.37 ± 13.58	73.93	0.132
7	SSS ^4^/SSS + LS, *n* (%)	45 (56.25%)	34.56	NA	NA	NA
8	LS ^5^, *n* (%)	23 (28.75%)	17.66	NA	NA	NA
9	CAV.S ^6^, *n* (%)	12 (15%)	9.21	NA	NA	NA
10	*PAI*^7^*5G/5G*, *n* (%)	6 (7.5%)	9.97	20 (25%)	16.02	*p* < 0.05
11	*PAI*^7^*4G/4G*, *n* (%)	32 (40%)	30.49	22 (27.5%)	23.5	*p* < 0.05
12	*PAI*^7^*4G/5G*, *n* (%)	42 (52.5%)	42.45	38 (47.5%)	37.52	0.245 *
13	tHcy ^8^, μmol/L	31.39 ± 31.25	28.03	12.3 ± 7.41	16.47	*p* < 0.05
14	TC ^9^, mg/dL	204.8 ± 36.60	210.5	198.26 ± 34.11	192.46	*p* < 0.05
15	LDLc ^10^, mg/dL	117.11 ± 25.89	118.96	108.11 ± 26.21	106.21	0.050
16	HDLc ^11^, mg/dL	56.65 ± 15.19	58.16	54.6 ± 12.14	53.06	0.242
17	TGL ^12^, mg/dL	144.15 ± 70.41	145.93	131.33 ± 67.73	129.37	*p* < 0.05
18	Blood glucose ^13^, mg/dL	102.07 ± 34.93	103.6	93.91 ± 24.56	92.33	0.084
19	HbA1c ^14^, %	5.56 ± 1.35	5.67	5.24 ± 0.95	5.12	0.703 *
20	hsCRP ^15^, mg/L	7.67 ± 2.94	7.06	4.39 ± 1.7	4.99	*p* < 0.05
21	IMT ^16^, mm	0.93 ± 0.17	0.95	0.91 ± 0.16	0.88	*p* < 0.05 *

* Fisher’s Test significant difference; Mann-Whitney U Test and Chi Square Test significant difference *p*-value < 0.05; *n* = absolute number of patients; % = relative number of patients; ^1^ HTN, hypertension; ^2^ SBP, systolic blood pressure; ^3^ DBP, diastolic blood pressure; ^4^ SSS, superior sagittal sinus; ^5^ LS, lateral sinus; ^6^ Cav. S, cavernous sinus; ^7^
*PAI-1*, plasminogen activator- inhibitor gene mutation (polymorphic variant); ^8^ tHcy, homocysteine; ^9^ TC, total cholesterol; ^10^ LDLc, low- density lipoprotein cholesterol; ^11^ HDLc, high-density lipoprotein cholesterol; ^12^ TGL, triglycerides; ^13^ Blood glucose, mg/dL; ^14^ HbA1c, glycohemoglobin; ^15^ hsCRP, high sensitive C-reactive protein; ^16^ IMT, intima-media thickness.

**Table 2 metabolites-11-00266-t002:** Comparison of homocysteine (tHcy) levels according to *PAI-1* gene polymorphism in the CVST group and controls group.

	CVST Group (*n* = 80)		Controls (*n* = 80)		
*4G/5G* Genotype	Number (%)	tHcy, µmol/L	Number (%)	tHcy, µmol/L	*p*-Value
		Mean ± Std		Mean ± Std	
*5G/5G*	6 (7.5%)	21.5 ± 11.67	20 (25%)	11.1 ± 3.17	0.001
*4G/4G*	32 (40%)	50.56 ± 37.98	22 (27.5%)	20.22 ± 7.41	0.032
*4G/5G*	42 (52.5%)	18.19 ± 17.16	38 (47.5%)	8.14 ± 4.94	0.516

## Data Availability

3rd Party Data. Restrictions apply to the availability of these data. Data was obtained from Timisoara County Emergency Clinical Hospital and are available from the authors with the permission of Institutional Ethics Committee of Clinical Studies of the Timisoara County Emergency Clinical Hospital.

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
