# Peer review of "Plasminogen Activator Inhibitor-1 (PAI-1) Gene Polymorphisms Associated with Cardiovascular Risk Factors Involved in Cerebral Venous Sinus Thrombosis"

_metabolites, 2021, doi:10.3390/metabo11050266_

Round 1
Reviewer 1 Report
Anca Elena Gogu et al. studied the potential association between PAI-1 675 4G/5G polymorphism with CVST risk and homocysteine (tHcy) level. The results indicate that 4G/4G genotype is associated with higher tHcy level in both control and CVST groups. The results are interesting, however, some claims the authors made may need adjustments to better fit their results, and additional analysis of the current data may provide more information regarding the relationship between PAI-1 4G/5G polymorphism and CVST risk. Here are specific points:
- In the abstract the authors wrote: “The PAI-1 675 4G/5G gene polymorphisms was significantly correlated with… higher total cholesterol (TC) (p<0.05)… in patients with CVST when compared with controls.” It sounds like that those cardiovascular risk factors are correlated with PAI-1 675 4G/5G polymorphism. However those factors in 4G/4G, 4G/5G, and 5G/5G patients were not directly compared. Similarly, from line 309: “The PAI-1gene mutation, especially the prevalence of 4G/4G genotype was correlated with higher total cholesterol (TC), low-density lipoprotein cholesterol (LDLc), triglycerides (TGL), homocysteine level (tHcy) and high-sensitive C-reactive protein (hsCRP) in our study.”, while there are no direct comparisons for those factors between 4G/4G patients from each group.
- Though CVST 4G/4G patients have higher tHcy than 4G/4G control patients, since the level of tHcy is already high in overall CVST patients, more comparisons on clinical features and cardiovascular factor measurements need to be compared between CVST and control patients with 4G/4G, 4G/5G, and 5G/5G genotype.
- Based on table 2 it is hard to make the conclusion that tHcy level is also high in 4G/5G CVST patients than 4G/5G control patients (line 240), since the difference is not significant.
- The authors should provide more detailed information on the methods of measurement. For example, detailed method for PAI-1 675 4G/5G polymorphism PCR needs to be shown.
- For clarity, it is suggested that “group 1” and “group 2” in the text should be replaced by “CVST group” and “control group”.
Author Response
Thank you very much for your help and guidance!
Anca Elena Gogu et al. studied the potential association between PAI-1 675 4G/5G polymorphism with CVST risk and homocysteine (tHcy) level. The results indicate that 4G/4G genotype is associated with higher tHcy level in both control and CVST groups. The results are interesting, however, some claims the authors made may need adjustments to better fit their results, and additional analysis of the current data may provide more information regarding the relationship between PAI-1 4G/5G polymorphism and CVST risk. Here are specific points:
- In the abstract the authors wrote: “The PAI-1 675 4G/5G gene polymorphisms was significantly correlated with… higher total cholesterol (TC) (p<0.05)… in patients with CVST when compared with controls.” It sounds like that those cardiovascular risk factors are correlated with PAI-1 675 4G/5G polymorphism. However those factors in 4G/4G, 4G/5G, and 5G/5G patients were not directly compared. Similarly, from line 309: “The PAI-1gene mutation, especially the prevalence of 4G/4G genotype was correlated with higher total cholesterol (TC), low-density lipoprotein cholesterol (LDLc), triglycerides (TGL), homocysteine level (tHcy) and high-sensitive C-reactive protein (hsCRP) in our study.”, while there are no direct comparisons for those factors between 4G/4G patients from each group.
The study was updated with the following comparisons, following the Reviewer’s advice:
Lines 33-36: From the PAI-1 gene polymorphisms, the PAI-1 675 4G/5G gene polymorphism presented statistically significant values regarding the comparisons of the blood lipids values between the CVST group and control group.
Lines: 305-367 “In our comparison of the CVST group and control group taking into consideration the PAI-1 gene polymorphism, regarding the levels of the TC, LDLc, HDLc, TGL and hsCRP levels we have found statistical significance regarding the following:
- in the 5G/5G comparison was a statistically significant difference between the TGL levels (p<0.001) between the CVST group and control group. The differences between the TC levels were not statistically significant (195.5 vs 194.2; p= 0.712) between the CVST group and control group. The differences between the LDLc levels were not statistically sig-nificant (116 vs 110.4; p= 0.124) between the CVST group and control group. The dif-ferences between the hsCRP levels were not statistically significant (6.61 vs 4.34; p= 0.398) between the CVST group and control group.
- in the 4G/4G comparison was not found a statistically significant difference between the TC, LDLc, TGL and hsCRP levels between the CVST group and control group. The differences between the TC levels were not statistically significant (199.03 vs 203.68; p= 0.201) between the CVST group and control group. The differences between the LDLc levels were not statistically significant (116.03 vs 114.23; p=0.571) between the CVST group and control group. The differences between the TGL levels were not statistically significant (149.78 vs 153.32; p= 0.264) between the CVST group and control group. The differences between the hsCRP levels were not statistically significant (7.42 vs 4.84; p= 0.571) between the CVST group and control group.
- in the 4G/5G comparison was a statistically significant difference between the TC, LDLc and TGL levels (p<0.001) between the CVST group and control group. The differences between the hsCRP levels were not statistically significant (8.02 vs 4.17; p= 0.264) between the CVST group and control group.”
- Though CVST 4G/4G patients have higher tHcy than 4G/4G control patients, since the level of tHcy is already high in overall CVST patients, more comparisons on clinical features and cardiovascular factor measurements need to be compared between CVST and control patients with 4G/4G, 4G/5G, and 5G/5G genotype.
The study was updated with the following comparisons, following the Reviewer’s advice:
Lines: 305-367 “In our comparison of the CVST group and control group taking into consideration the PAI-1 gene polymorphism, regarding the levels of the TC, LDLc, HDLc, TGL and hsCRP levels we have found statistical significance regarding the following:
- in the 5G/5G comparison was a statistically significant difference between the TGL levels (p<0.001) between the CVST group and control group. The differences between the TC levels were not statistically significant (195.5 vs 194.2; p= 0.712) between the CVST group and control group. The differences between the LDLc levels were not statistically sig-nificant (116 vs 110.4; p= 0.124) between the CVST group and control group. The dif-ferences between the hsCRP levels were not statistically significant (6.61 vs 4.34; p= 0.398) between the CVST group and control group.
- in the 4G/4G comparison was not found a statistically significant difference between the TC, LDLc, TGL and hsCRP levels between the CVST group and control group. The differences between the TC levels were not statistically significant (199.03 vs 203.68; p= 0.201) between the CVST group and control group. The differences between the LDLc levels were not statistically significant (116.03 vs 114.23; p=0.571) between the CVST group and control group. The differences between the TGL levels were not statistically significant (149.78 vs 153.32; p= 0.264) between the CVST group and control group. The differences between the hsCRP levels were not statistically significant (7.42 vs 4.84; p= 0.571) between the CVST group and control group.
- in the 4G/5G comparison was a statistically significant difference between the TC, LDLc and TGL levels (p<0.001) between the CVST group and control group. The differences between the hsCRP levels were not statistically significant (8.02 vs 4.17; p= 0.264) between the CVST group and control group.”
- Based on table 2 it is hard to make the conclusion that tHcy level is also high in 4G/5G CVST patients than 4G/5G control patients (line 240), since the difference is not significant.
Yes, indeed, the text was modified (line 300).
- The authors should provide more detailed information on the methods of measurement. For example, detailed method for PAI-1 675 4G/5G polymorphism PCR needs to be shown.
The procedure was explained and updated in the text lines: 114-118.
“The QIAamp DNA Blood Mini Kit facilitates the segregation of DNA from blood with fast spin-column or vacuum processes. The QIAamp DNA Blood Mini Kit operates sample sizes of up to 200 µl, with a preparation time of 20–40 minutes. The usual yield from 200 µl healthy whole blood is 4–12 µg, with an elution volume of 50–200 µl. The QIAamp DNA Blood Mini Kit process is automatable.”
- For clarity, it is suggested that “group 1” and “group 2” in the text should be replaced by “CVST group” and “control group”.
The replacement was done, thank you for the suggestion!
Reviewer 2 Report
This manuscript is interesting, but the following points need to be revised.
1Neuroimaging CT and MRI images need to be supplemented in the manuscript.
2 The study patient population of this manuscript is mainly young patients. How old is the patient reported in the previous literature? What is the innovativeness of this manuscript?
3 Table 2 Comparison of homocysteine (tHcy) levels according to 4G/5G polymorphism in the patients and controls group. Why not make other similar comparisons such as LDLc, HDLc, etc.?
5 Why is tHay different in Table 1 and Table 2? It should be explained.
6 There are too many descriptions of repeated results in the discussion and need to be carefully revised.
This manuscript is interesting, but the following points need to be revised.
1Neuroimaging CT and MRI images need to be supplemented in the manuscript.
2 The study patient population of this manuscript is mainly young patients. How old is the patient reported in the previous literature? What is the innovativeness of this manuscript?
3 Table 2 Comparison of homocysteine (tHcy) levels according to 4G/5G polymorphism in the patients and controls group. Why not make other similar comparisons such as LDLc, HDLc, etc.?
5 Why is tHay different in Table 1 and Table 2? It should be explained.
6 There are too many descriptions of repeated results in the discussion and need to be carefully revised.
This manuscript is interesting, but the following points need to be revised.
1Neuroimaging CT and MRI images need to be supplemented in the manuscript.
2 The study patient population of this manuscript is mainly young patients. How old is the patient reported in the previous literature? What is the innovativeness of this manuscript?
3 Table 2 Comparison of homocysteine (tHcy) levels according to 4G/5G polymorphism in the patients and controls group. Why not make other similar comparisons such as LDLc, HDLc, etc.?
5 Why is tHay different in Table 1 and Table 2? It should be explained.
6 There are too many descriptions of repeated results in the discussion and need to be carefully revised.
This manuscript is interesting, but the following points need to be revised.
1Neuroimaging CT and MRI images need to be supplemented in the manuscript.
2 The study patient population of this manuscript is mainly young patients. How old is the patient reported in the previous literature? What is the innovativeness of this manuscript?
3 Table 2 Comparison of homocysteine (tHcy) levels according to 4G/5G polymorphism in the patients and controls group. Why not make other similar comparisons such as LDLc, HDLc, etc.?
5 Why is tHay different in Table 1 and Table 2? It should be explained.
6 There are too many descriptions of repeated results in the discussion and need to be carefully revised.
This manuscript is interesting, but the following points need to be revised.
1Neuroimaging CT and MRI images need to be supplemented in the manuscript.
2 The study patient population of this manuscript is mainly young patients. How old is the patient reported in the previous literature? What is the innovativeness of this manuscript?
3 Table 2 Comparison of homocysteine (tHcy) levels according to 4G/5G polymorphism in the patients and controls group. Why not make other similar comparisons such as LDLc, HDLc, etc.?
5 Why is tHay different in Table 1 and Table 2? It should be explained.
6 There are too many descriptions of repeated results in the discussion and need to be carefully revised.
This manuscript is interesting, but the following points need to be revised.
1Neuroimaging CT and MRI images need to be supplemented in the manuscript.
2 The study patient population of this manuscript is mainly young patients. How old is the patient reported in the previous literature? What is the innovativeness of this manuscript?
3 Table 2 Comparison of homocysteine (tHcy) levels according to 4G/5G polymorphism in the patients and controls group. Why not make other similar comparisons such as LDLc, HDLc, etc.?
5 Why is tHay different in Table 1 and Table 2? It should be explained.
6 There are too many descriptions of repeated results in the discussion and need to be carefully revised.
Author Response
Thank you very much for your help and guidance!
This manuscript is interesting, but the following points need to be revised.
1Neuroimaging CT and MRI images need to be supplemented in the manuscript.
The neuroimaging chapter was updated with representative images from our patients.
2 The study patient population of this manuscript is mainly young patients. How old is the patient reported in the previous literature? What is the innovativeness of this manuscript?
In our study were included mainly young patients, with the ages over 18 years old because our Neurology Clinic includes only adults, it is not a paediatric Neurology Clinic. Also, the adults over 45 years presented many comorbidities which could affect the precision and accuracy of our study; therefore, they were excluded. In the study of Akhter et al [36], the participants were young adults too (18-45 years).
“PAI and homocysteine are involved in neurological brain pathology and Marcurcci et al [21] wrote about the occurrence of major adverse cardiac events after successful coronary stenting as being related to PAI-1 and homocysteine. (line 521)”
The innovativeness of the study comes from the fact that this study aims to broaden the primary diagnostic horizon in young adults in Southeast Europe to consider blood aggregability and genetic polymorphisms as both causative and risk factors in neurological vascular pathology. In particular, in Romania, it is necessary to consider these diagnoses and perform further studies to diagnose the causative factors, but also the risk in vascular neurological diseases.
3 Table 2 Comparison of homocysteine (tHcy) levels according to 4G/5G polymorphism in the patients and controls group. Why not make other similar comparisons such as LDLc, HDLc, etc.?
The study was updated with the following comparisons, following the Reviewer’s advice:
Lines: 305-367 “In our comparison of the CVST group and control group taking into consideration the PAI-1 gene polymorphism, regarding the levels of the TC, LDLc, HDLc, TGL and hsCRP levels we have found statistical significance regarding the following:
- in the 5G/5G comparison was a statistically significant difference between the TGL levels (p<0.001) between the CVST group and control group. The differences between the TC levels were not statistically significant (195.5 vs 194.2; p= 0.712) between the CVST group and control group. The differences between the LDLc levels were not statistically sig-nificant (116 vs 110.4; p= 0.124) between the CVST group and control group. The dif-ferences between the hsCRP levels were not statistically significant (6.61 vs 4.34; p= 0.398) between the CVST group and control group.
- in the 4G/4G comparison was not found a statistically significant difference between the TC, LDLc, TGL and hsCRP levels between the CVST group and control group. The differences between the TC levels were not statistically significant (199.03 vs 203.68; p= 0.201) between the CVST group and control group. The differences between the LDLc levels were not statistically significant (116.03 vs 114.23; p=0.571) between the CVST group and control group. The differences between the TGL levels were not statistically significant (149.78 vs 153.32; p= 0.264) between the CVST group and control group. The differences between the hsCRP levels were not statistically significant (7.42 vs 4.84; p= 0.571) between the CVST group and control group.
- in the 4G/5G comparison was a statistically significant difference between the TC, LDLc and TGL levels (p<0.001) between the CVST group and control group. The differences between the hsCRP levels were not statistically significant (8.02 vs 4.17; p= 0.264) between the CVST group and control group.”
5 Why is tHay different in Table 1 and Table 2? It should be explained.
The median of tHcy levels in the first table is the total value for all patients with genetic changes.
The median in the second table is determined by genotype. We also wanted to find out if there was any concordance between genotypes.
Therefore, the median is higher than normal, but lower than in the first table.
6 There are too many descriptions of repeated results in the discussion and need to be carefully revised.
The text of the Discussions chapter was updated.
This manuscript is a resubmission of an earlier submission. The following is a list of the peer review reports and author responses from that submission.